# Time Preferences between Individuals and Groups in the Transition from Hunter-Gatherer to Industrial Societies

**Yayan Hernuryadin** [1,2] **, Koji Kotani** [1,3,4,5,] ***** **and Yoshio Kamijo** [1,3]

[1]  School of Economics and Management, Kochi University of Technology, Kami 782-0003, Japan; boyan_nuryadin@yahoo.co.id (Y.H.); yoshio.kamijo@gmail.com (Y.K.)
[2]  Ministry of Marine Affairs and Fisheries, Medan Merdeka Timur No. 16, Jakarta 10110, Indonesia
[3]  Research Institute for Future Design, Kochi University of Technology, Kami 782-0003, Japan
[4]  Urban Institute, Kyusyu University, Fukuoka 819-0395, Japan
[5]  College of Business, Rikkyo University, Tokyo 171-8501, Japan
*  Correspondence: kojikotani757@gmail.com

**Abstract:** Three societies, namely the hunter-gatherer, the agrarian and the industrial, represent the course of human history of cultural and economic development. In this course, each society exhibits distinct cultures and daily life practices that shape human behaviors and preferences, characterizing temporal actions and consequences at the individual and group levels. We examine individual and group time preferences and their relation across the three societies. To this end, we conduct a field experiment to elicit individual and group discount factors in three societies of Indonesia—(i) the fisheries, (ii) the farming and (iii) the urban societies—as proxies of the hunter-gatherer, agrarian and industrial societies, respectively. We find that both individual and group discount factors are the lowest (highest) in the fisheries (agrarian) society, while those in the urban society are in the middle. We also observe that the determinants of group discount factors differ across societies: members of the lowest and middle discount factors in a group play an important role in determining the group discount factor in the fisheries society, while only the members with the middle discount factor are key in agrarian and urban societies. Overall, our results suggest that individual and group discount factors non-monotonically change as societies transition from fisheries to agrarian and from agrarian to urban and that comparatively shortsighted people (the lowest and middle) are more influential than farsighted people in determining group time preferences.

**Keywords:** discount factors; individuals and groups; fisheries society; farming society; urban society

## 1. Introduction

Three societies, namely the hunter-gatherer, the agrarian and the urban, have shaped the course of human history through economic and cultural development [1]. In this course, each society exhibits distinct cultures and daily life practices that characterize temporal actions and consequences at individual and group levels. Shahrier et al., Ma et al., Shahrier et al., and Timilsina et al. [2–5] suggest that a transition of societies from rural to urban affects social preferences and behaviors. Moreover, such changes in preferences and behaviors are claimed to be related to people's temporal actions and consequences at the individual and group levels. For example, Indonesian fishermen work in a group to spot fishing grounds and catch fish in a competitive and harsh environment; farmers coordinate their efforts with other farmers for irrigation, planting, growing, and harvesting in uncertain climate conditions; and urban people live or work in an environment that is surrounded by technologies and

detached from nature. This paper addresses individual and group time preferences as well as their relation across different societies.

Several works have examined how sociodemographic and environmental factors characterize time preferences [6–13]. Tanaka et al., Harrison et al., and Reimers et al. [6,10,13] demonstrate that age, income, and education are correlated with time preferences. Another group of researchers show that individual time preferences can be explained by environments and occupations. Nguyen [12] presents that fishermen with experiences of participating in resource conservation programs are more future-oriented than those with other occupations. Johnson and Saunders [11] demonstrate that divers are more future-oriented than fishermen since divers are required to be patient to maintain healthy ocean for sustainability in their daily occupation. In addition, Casse and Nielsen, and Duquette et al. [7,8] examine farmers' time preferences and find that farmers with more future-oriented preferences tend to adopt the best management practices in earlier stages or never perform slash-and-burn agriculture. Galor and Ozak [9] demonstrate that people evolve to have long-term orientations when they have lived in a region where high return to agricultural investment or crop yield is expected. Some studies, such as Duncan et al., Ekeland et al., and Da-Rocha et al. [14–16] suggest that time preferences of resource users are characterized by non-constant discount factors, demonstrating that such non-constant discount factors of resource users significantly affect harvesting strategies and the associated trajectories of resource stocks.

The relationships between individual and group time preferences have been studied by several researchers. Charlton et al., Gillet et al., Sutter, and Denant-Boemont et al. [17–20] show that people tend to be more impatient in individual decisions than they are in group decisions. However, Yang and Carlsson [21] find that individual decisions are similar to joint decisions in terms of time preferences. Another group of works examine time preferences and social preferences, finding that more patient subjects are likely to share payoffs with other people in social-dilemma situations [3,22,23]. Ambrus et al., and He and Villeval [24,25] demonstrate that a "median" member (who has a median social preference in a group) has a significant influence on group decisions since the highest and the lowest members tend to be attracted to the median.

None of the past studies have addressed individual and group time preferences, focusing on the transition of societies in cultural and economic development. We examine individual and group time preferences as well as their relation across hunter-gatherer, agrarian and industrial societies, reflecting the course of human history. To this end, we conduct a field experiment regarding individual and group discount factors for three societies of Indonesia, (i) fisheries, (ii) farming and (iii) urban, as proxies of hunter-gatherer, agrarian and industrial societies, respectively. (Some literature characterizes fisheries societies as hunter-gatherer societies because of their daily life practices [26,27].) Our empirical analysis yields two main results. First, we find that both individual and group discount factors are the lowest (highest) in fisheries (agrarian) societies, while those in urban ones are in the middle. Second, we observe that the determinants of group discount factors differ across the three societies; members with the lowest and middle discount factors in a group play an important role in making a group discount factor in fisheries societies, while only the member with the middle discount factor is key in agrarian and urban societies. Overall, our results imply that individual and group discount factors non-monotonically shift as societies change from fisheries to agrarian and from agrarian to urban and that comparatively shortsighted people (the lowest and middle) are more influential than farsighted people in determining group time preferences.

## 2. Methods and Materials

### 2.1. Study Areas

The questionnaire surveys and experiments were conducted in Karawang and Jakarta with three different societies: fisheries and agrarian villages in Karawang and an urban city in Jakarta (Figure 1). Karawang regency is in the north part of Jawa Barat Province. Karawang is located between $107°2'$

and 107°40′ east longitude, and 5°56′ and 6°34′ south latitude. The population in 2015 was 2,273,579 with a density of 1094 km² [28], and 168,901, or 18.15 % of the working population, work in agriculture and fishery sectors [29]. Jakarta is the most densely populated and capital city in Indonesia, where most people engage in the government, business, and service sectors. Jakarta is located at 6°12′ S and 106°48′ E. The population in 2016 was 10,277,628 with a density of 15,517 km², and 3,136,531, or 64.51 % of the working population, work as regular employees in the public and formal private sectors [30].

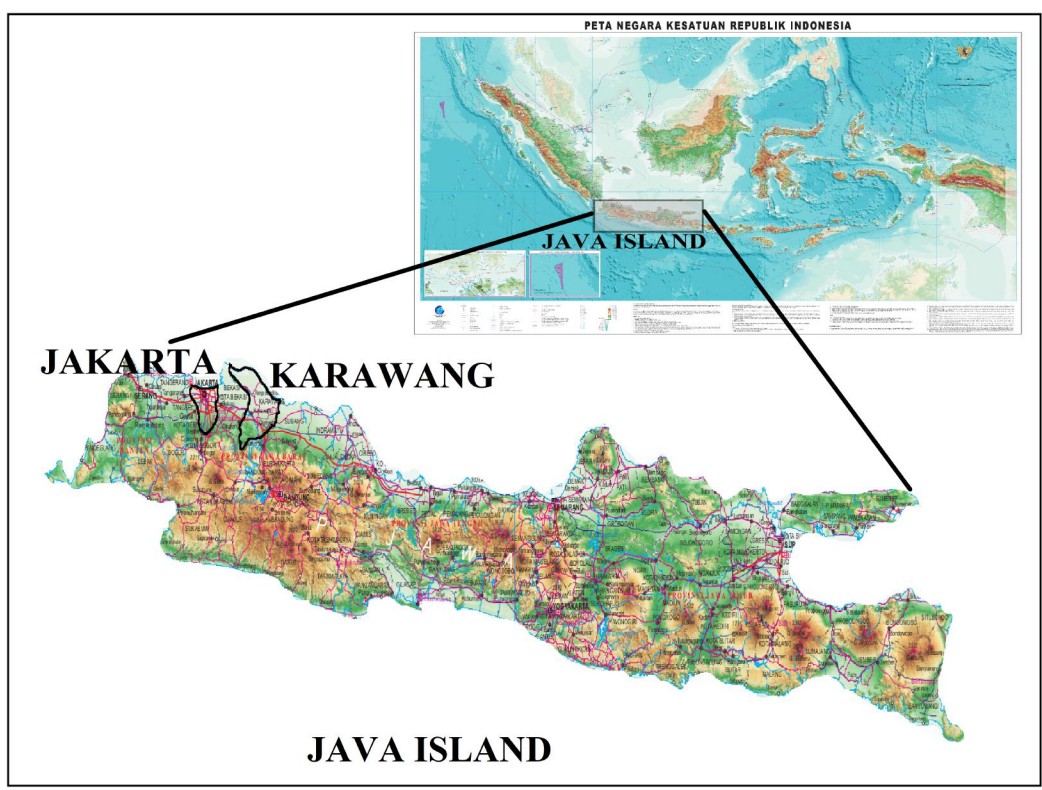

**Figure 1.** The study area: Karawang and Jakarta.

## 2.2. Experimental Setup

### 2.2.1. A Discounting Elicitation Experiment

We employ a discounting elicitation experiment to identify individual and group time preferences. Our experiment is different from a multiple-price list (MPL) procedure as done by [6,10,31]. Subjects in MPL are given payoff tables with many questions including the information of interest rates. The MPL procedure requires subjects to have bank accounts, enabling them to receive experimental reward at later dates. (We initially considered application of the MPL procedure, because most field experiments that seek to estimate time preferences have employed it [6,11,12,32].) However, most of our subjects are local farmers and fishermen who do not have bank accounts. Also, such subjects claim that they could neither understand nor follow the MPL procedure in the pilot experiments, because the procedures do not match their lifestyle and their educations. (Many subjects in the MPL pilot experiments randomly answer or have difficulty understanding the questions. The same type of problems and high inconsistency that comes from respondents' random answers are reported in literature [8,33–35].) Therefore, we design and institute a simple discounting elicitation experiment that consists of face-to-face interviews to ask subjects to choose between receiving money today and more money one month later. We employ a face-to-face interview under complete privacy for discounting elicitation because the method is reported to create an environment of trust and confidentiality and enables us to effectively elicit people's true thoughts and preferences, even when subjects are not

fully motivated to express what they think and prefer by monetary incentive or when the scenario is explained to be hypothetical in experiments [36–42].

We conduct the discounting elicitation experiments for each subject and a group of 3 subjects to identify individual and group discount factors. First, we announce that our elicitation experiments are conducted with public support and approval from Indonesia government and local community authorities, and thus, 20,000 Rp ($\approx$1.50 USD) of experimental rewards would be paid to each subject and to each group on an average, respectively, as far as subjects and groups honestly and truthfully answer the questions in a face-to-face interview, reflecting their daily money senses and life. (All groups decide to split the group experimental rewards into three.) In this announcement, we do not purposely detail how experimental payments shall be made to the subjects at a specific time and date. Second, we start eliciting individual time preferences through a discounting elicitation experiment where subjects are asked to answer a series of questions. As most subjects are not well educated and have limited literacy, we institute simple experiments and instructions that our subjects can understand.

In a face-to-face interview in the discounting elicitation experiments, an interviewer begins by asking a question of whether each subject would choose option $A$ or $B$ with complete privacy of a separate room in the following scenario:

Option $A$: You get 20,000 Rp today. (1 USD $\approx$ 13,350 Rp in January 2017.)
Option $B$: You get 20,000 + $m$ Rp one month later.

The value of $m$ in option $B$ begins with $m_0 = 4000$. When the subject chooses option $A$, we proceed to the next question in which the value of $m$ is increased by 4000, i.e., $m = m_1 = m_0 + 1 \cdot 4000 = 4000 + 4000 = 8000$. Then, the subject is asked whether she prefers option $A$ or $B$. As long as the subject continues to select option $A$, the experiment continues, with the value of $m$ for option $B$ increasing by 4000. This updating procedure for $m$ in option $B$, i.e., $m_k = 4000 + k \cdot 4000$, continues an arbitrary $k$ times for as long as a subject prefers option $A$ to $B$. We shall end the update process when the subject chooses option $B$ for the first time at the $k + 1$th question where the value of $m$ in option $B$ is updated with $m_{k+1} = m_0 + (k + 1) \cdot 4000$. In this case, we consider that her preference over options $A$ and $B$ is reversed between $k$th and $k + 1$th questions, and there should exist a threshold future value of $\overline{m}$ between $m_k$ and $m_{k+1}$ that makes the subject to be indifferent between receiving 20,000 Rp today and 20,000 $+\overline{m}$ Rp one month later. Therefore, as a final process, we interview the subject and ask her some final questions by gradually adjusting the value of $m$ between $m_k$ and $m_{k+1}$ up until each interviewer identifies the threshold value of $\overline{m}$. The subject's individual discount factor shall be estimated to be $\rho = \frac{20,000}{20,000+\overline{m}}$, which follows a definition of discount factors introduced in Sanni et al., and Smith [43,44]. Sanni et al., and Smith [43,44] define a discount factor to be the present value of one unit of currency at some future date. Since our focus is on time preferences, not on curvatures of utility functions, we follow the simple definition of discount factors in this field experiment without assuming any utility function.

After completing a discounting elicitation game at the individual level, we proceed to the experiment at the group level. We randomly choose 3 subjects and assign them to a group. We then implement the same procedures as we did at the individual level and ask the group whether to choose option $A$ or $B$. The difference at the group level is that the decisions between options $A$ and $B$ in each trial must be discussed among group members. We ask group members to reach a consensus through discussion for every group decision without relying on majority voting. When the group chooses option $B$ for the first time at the $k + 1$th question, we end the updating process and ask the group a series of questions to identify the group's threshold value of $\overline{m}$ that makes the group to be indifferent between receiving 20,000 Rp today and 20,000 $+\overline{m}$ Rp one month later. Therefore, the group's discount factor shall be estimated in the same way we did for individual discount factors.

To confirm whether identified $\overline{m}$s in discounting elicitation at individual and group levels are within a plausible range, we finally announce and prepare a lottery game where a certain option

and a probabilistic option are given with the same expected payoff at the end of each session. After both individual and group discounting elicitation experiments finish, we announce that a lottery game is prepared and implemented based on individual and group $\overline{m}$s. (Subjects come to know, for the first time, an existence of the lottery game after individual and group discounting elicitation experiments. Therefore, $\overline{m}$s identified through discounting elicitation are independent of the lottery games.) In the lottery game, 20 cards are yellow and $\frac{\overline{m}}{1000}$ cards are red, these cards are counted in front of subjects (or groups) and put into a bag. (Most subjects do not understand the concept of probabilities. Therefore, we count the number of red and yellow cards in front of them.) In this lottery, when a subject (or group) picks a yellow card, she receives the reward of 20,000 $+\overline{m}$ Rp, otherwise zero. Mathematically, the lottery has a probability $\rho = \frac{20,000}{20,000+\overline{m}}$ of successfully receiving the value of 20,000 $+\overline{m}$ Rp by picking a yellow card and a probability $1 - \rho$ of receiving nothing by picking a red card with the expected payoff of 20,000 Rp.

After counting the cards in front of subjects and setting them in the bag, we explain and ask each subject (each group) to choose between certainly receiving 20,000 Rp (a certain option) and going for the lottery to possibly receive 20,000 $+\overline{m}$ Rp (a probabilistic option). A subject (group) who chooses the lottery receives the payment according to the outcome of a random draw from the bag; a subject (group) who does not choose the lottery certainly receives 20,000 Rp. (Regarding the payment to each group, a group is asked to discuss and decide between a certain option and a probabilistic option as well as how to split the payment among themselves. The total payment each subject receives from the experiments is the sum of the payments from individual elicitation and the split from group one.) The $\rho$s that we have identified in our experiment could be considered appropriate as probabilities for our lotteries even under expected utility framework in choice under uncertainty (see, e.g., Mas-Colell et al. [45]). Suppose there are two utility levels, $u(20,000)$ and $u(20,000+\overline{m})$, correspond to the payoff of 20,000 and 20,000 $+\overline{m}$ for each subject. If our discounting elicitation is reasonable enough for subjects to truthfully reveal $\overline{m}$s and the associated $\rho$s following Sanni et al., and Smith [43,44], subjects in our experiment have two choices in the lottery with the equal expected payoff of 20,000: (1) a certain option in which the subject obtains an expected utility of $u(20,000)$ and (2) a probabilistic option in which she obtains an expected utility of $\rho \cdot u(20,000+\overline{m})$.

Past literature consistently demonstrates that approximately 40–60% of subjects prefer to choose a certain option to a probabilistic option in lottery games under "reasonable" support with the same expected payoff [46–52]. Also, some other literature shows that when the support of a lottery game is highly skewed and asymmetric (e.g., huge return with tiny probability such as buying lottery tickets), most subjects prefer to choose a probabilistic option to a certain option [53–55]. Such a skewed and asymmetric lottery corresponds to the cases in our lottery where $\overline{m}$s are reported to be unreasonably high by subjects or groups (or corresponding $\rho$s become so small) especially when they tell lies or keep saying that option *A* is preferred to option *B*. We consider that approximately 40–60% of our subjects should choose a certain option in our lottery game as far as the identified $\overline{m}$s in our elicitation are within a plausible range, being consistent with the previous results in lottery games under reasonable supports. Otherwise, most of our subjects might choose a probabilistic option, being consistent with the results in skewed and asymmetric lotteries.

In fact, we conducted the pilot experiments with 30 subjects and 10 groups in each society consisting of discounting elicitation and lottery procedures. We confirmed that subjects and groups understood what we ask in a face-to-face interview to reveal $\overline{m}$s. In particular, it appeared that they had neither incentives to tell a lie nor intentions for higher $\overline{m}$s by continuously saying that option *A* is preferred to option *B* in discounting elicitation. This may be because we successfully established a private space for each subject and each group in discounting elicitation and subjects trust us by noting that our experiments are conducted under public support and consent from Indonesia government and local community authorities. After discounting elicitation in the pilot experiments, we announced and implemented a lottery game based on $\overline{m}$s by asking subjects (groups) to choose between a certain option and a probabilistic option. The result was that 45 % (43 %) of subjects (groups) chose a certain

option, being consistent with past literature of lottery games under reasonable support. Therefore, we judge that subjects neither tell a lie nor have any specific incentive to manipulate $\overline{m}$s in elicitation and decide to implement a current experimental design.

### 2.2.2. Social Value Orientation Games

We use the social value orientation (SVO) game suggested by Murphy et al. [56] to measure subjects' social preferences. This method categorizes an individual value orientation into altruism, pro-sociality, individualism, or competitiveness depending upon their choices in the SVO game. In this game, subjects are asked to choose among nine options for each of six primary questions (See Figure 2). Subjects are randomly paired such that subjects do not know each other. Each question consists of a problem in which a subject decides to allocate points to herself and to the other subject in her pair by choosing one of nine options. After each subject has made her choices in all six questions, she is asked to write the resulting distributions of money between herself and the other subject on the spaces provided on the right-hand side of the SVO instruction sheets (Figure 2).

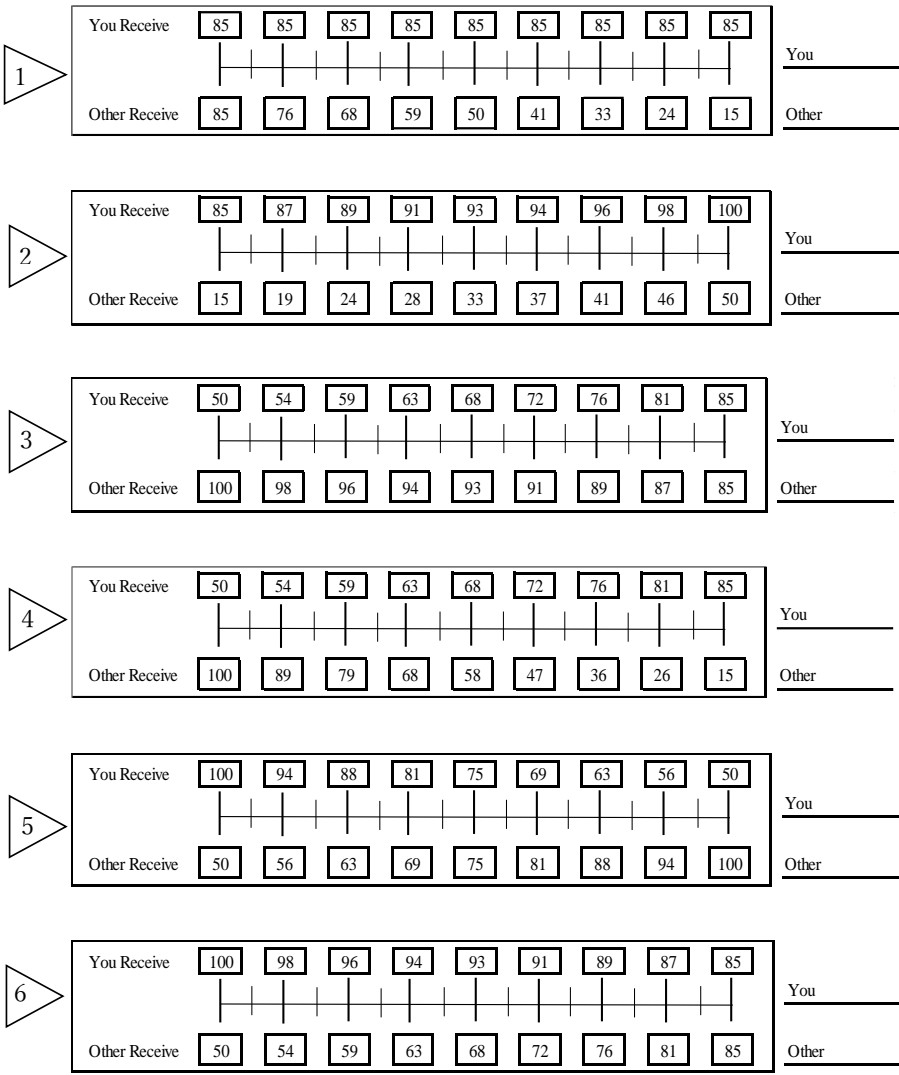

**Figure 2.** A social value orientation (SVO) game developed by Murphy et al. [56].

Subjects are informed that they get paid based on their earnings from the SVO game in the following manner. The total amount of points a subject is allocated by herself and by the other in her pair are calculated by summing the points from all 12 items (six items from each person in a pair).

The points are converted into real money with an experimental exchange rate. In our experiment, 1 point is equivalent to 200 Rp. The average payment in the SVO game is 28,000 Rp (approximately 2.10 USD). After the game, we identify a subject's SVO by computing the mean allocations for oneself $\bar{A}_s$ and for the other $\bar{A}_o$, from the choices on the six items. Then, 50 is subtracted from each of $\bar{A}_s$ and $\bar{A}_o$, and the inverse tangent of the ratio between $\bar{A}_s - 50$ and $\bar{A}_o - 50$ is calculated as the SVO angle, i.e., SVO $= \arctan \frac{\bar{A}_o - 50}{\bar{A}_s - 50}$. The subject is identified as an altruist if her SVO angle is greater than $57.15°$, prosocial if the angle is between $57.15°$ and $22.45°$, individualist if the angle is between $22.45°$ and $-12.04°$, and competitive if the angle is less than $-12.04°$.

## 2.3. Experimental Procedures

We implemented field experiments and surveys by employing different approaches of random sampling to fisheries and agrarian societies in Karawang and to urban societies in Jakarta because they have different economic and sociodemographic characteristics. In Karawang, we first contacted the local government office to get approval to conduct field research, and 3 fisheries and 9 agrarian villages gave us approval. We obtained a list of residents from their local government offices and randomly selected the required number of households based on the population of each village. Subsequently, we invited an income-earning member from each household to participate in our experiments by sending them an invitation letter. In total, 200 fishermen and 197 farmers participated in our field research.

In Jakarta, we randomly chose subjects based on occupations. First, we collected information about the proportion of each occupational category in the total population of the Jakarta area by referring to the BPS-Statistics of DKI (Daerah Khusus Ibukota) Jakarta Province. Then, we randomly selected several organizations or companies for each category and contacted their office to get approval to conduct our field research. We invited individuals from these companies and organizations based on their compliance. In total, 200 urban people participated in our field research, and the experiments were conducted at community halls in each area of Jakarta. Overall, 597 subjects participated in our experiment (197 farmers, 200 fishermen and 200 urban people). We asked each subject to leave the experimental site soon after completing all the tasks to prevent unnecessary interactions among subjects.

In each session of our field experiments, we prepared a printed experimental instruction (a discounting elicitation experiment and the SVO game) for subjects in the Indonesian language (Bahasa). The first author in this research explained the experimental procedures and rules by verbal presentation at a gathering room, and we also confirmed each subject's understanding by giving a series of quizzes about our experimental rules and procedures after the presentation. We first conducted the SVO games and then proceeded to the discounting elicitation experiments at the individual and group levels. In the discounting elicitation experiments, we randomly selected a subject and guided him/her to a separate room with complete privacy. We announced that 20,000 Rp ($\approx$1.50 USD) Rp shall be paid to each subject (each group) on an average, as far as subjects (groups) honestly and truthfully answer a series of questions and tasks in a face-to-face interview, reflecting their daily money senses and life. Before we started the discounting elicitation experiment, we clarified one more time whether subjects understood the procedure or not. We implemented the same procedures as we did at the individual level and ask the group whether to choose option *A* or option *B*. After completing a group discounting experiment, we conducted the lottery game to confirm whether $\overline{m}$ would be reasonable or not and to determine the experimental payments of discounting elicitation experiments. Finally, we conducted a field questionnaire survey to collect sociodemographic information each session. Each subject earned the average total experimental earnings of 70,000 Rp ($\approx$5.2 USD) from the SVO games, individual and group discounting elicitation experiments and the participation fee of 15,000 Rp ($\approx$1.1 USD). Figure 3 summarizes the experiment procedure for discounting elicitation. Approximately, 15–20 subjects participated in each session of our experiment, and each session took 3–4 h.

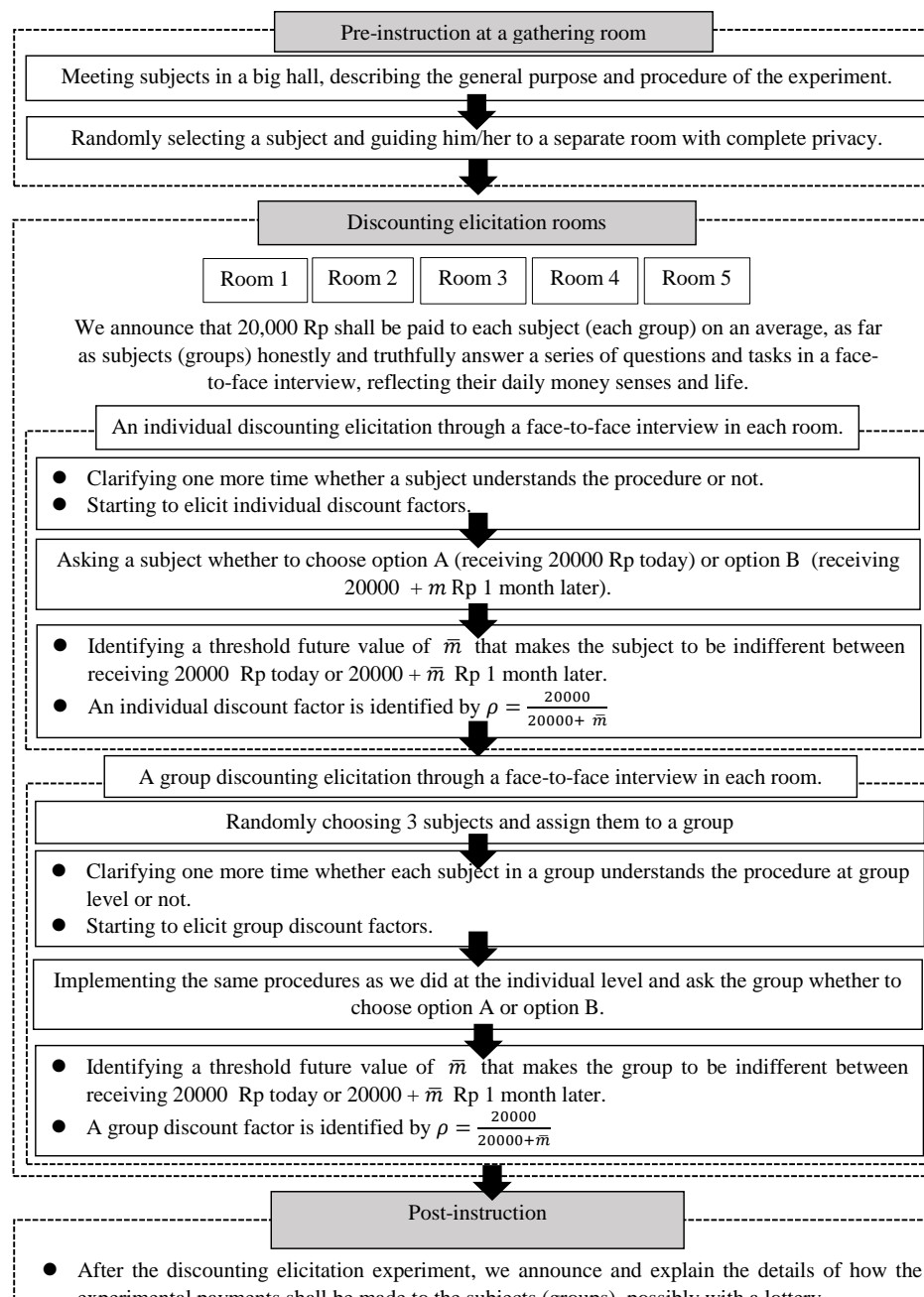

**Figure 3.** The experimental procedure for discounting elicitation.

*2.4. Empirical Method*

We employ betafit regression to identify factors that characterize group discount factors. Betafit models can be mathematically expressed as:

$$g_i = \beta_0 + \beta_1 \mathbf{x}_i + \beta_2 z_i + \epsilon_i, \tag{1}$$

wheresubscript $i$ represents each group's ID, $g_i$ is an estimated group discount factor, $\mathbf{x}_i$ is a vector of independent variables of categories of individual discount factors (the lowest, middle, and highest) and sociodemographic information, such as age, education, household income, number of household members, and occupation dummy. In addition, since the SVO categorizes individuals as altruist, prosocial, individualistic and competitive, and only 18 samples or 3.01 % of our data are identified as altruist or competitive, we merge the individualistic and competitive orientations into a "proself" category and merge altruist and prosocial into a "prosocial" category for simplicity of the analysis. Therefore, $z_i$ is a dummy variable of SVO that takes a value of 1 when subject $i$ is proself and is otherwise 0. $\beta_0$ ($\boldsymbol{\beta}_1$) and $\beta_2$ are the associated parameters (of vectors) to be estimated. Table 1 presents the definitions of the variables used in the regression analysis.

The betafit regression developed by Ferrari and Cribari-Neto [57] accommodates a group discount factor that is bounded between 0 and 1 as a dependent variable with the assumption that group discount factors $g_i$s follow a beta distribution:

$$f(g_i; \mu, \phi) = \frac{\Gamma(\phi)}{\Gamma(\mu\phi)\Gamma((1-\mu)\phi)} g_i^{\mu\phi-1} (1 - g_i)^{(1-\mu)\phi-1}, \quad g_i \in (0,1),$$

where $\mathbb{E}(g_i) = \mu$, $\text{Var}(g_i) = \frac{\mu(1-\mu)}{1+\phi}$, $\phi$ is an accuracy parameter and $\phi - 1$ is a distribution parameter. Various combinations of $\mu$ and $\phi$ determine the types of beta densities, such as $J$ shaped, inverted $J$ shaped and $U$ shaped [57]. The application of betafit regression appears to be valid because the distributions of the group discount factors estimated in our experiments are identified to be $U$ shaped and inverted $J$ shaped (see Figure 4a,b). The maximum likelihood method is used to determine the unknown parameters $\beta_0$, $\boldsymbol{\beta}_1$, and $\beta_2$ in Equation (1), with which the marginal effect of an independent variable on the group discount factors $g_i$s is obtained.

The variables in Table 1 are hypothesized to affect group discount factors. We rank the individual discount factors of 3 subjects in a group into the lowest, middle, and highest discount factors that are included as independent variables in Equation (1). A group discount factor is the elicited value of discounting the future value at the group level, as described in Section 2.2.1 and taken as a dependent variable in Equation (1). We are interested in how individual discount factors and the associated rankings affect group discount factors. The average age, income, and number of household members at the group level are also considered to affect group time preferences, following Harrison et al. [10] and Reimers et al. [13]. In addition, a number of proself members in a group are included in the models to capture how individual social preferences influence group time preferences. We define dummy variables for agrarian and urban societies, taking the fishery society as the reference group. The dummy variables are considered to see how a transition of societies from fisheries to farming and from farming to urban may have affected individual and group time preferences as well as their relations.

**Table 1.** Definitions of the variables used in the analysis.

| Variables | Description |
| --- | --- |
| Individual discount factor | Percentage rate of discounting the future monetary value that will definitely be received one month later in such a way that the discounted future value equals the value of receiving 20,000 Rp today. |
| Lowest individual discount factor | The individual discount factor that is the lowest among the three members in a group. |
| Middle individual discount factor | The individual discount factor that is the middle among the three members in a group. |
| Highest individual discount factor | The individual discount factor that is the highest among the three members in a group. |
| Group discount factor | Percentage rate of discounting the future monetary value as a group of three people that will definitely be received one month later in such a way that the discounted future value equals the value of receiving 20,000 Rp today. |
| Age | Average age of members in a group. |
| Household income | Average household income of group members per month in 1 million rupiahs. |
| Household members | Average number of household members in a group. |
| Number of proself members | Number of proself members in a group. |
| Society dummy variables (The reference = the fisheries) | |
| Agrarian dummy | Takes a value of one when the group of three people is in the agrarian society, otherwise zero. |
| Urban dummy | Takes a value of one when the group of three people is in the urban society, otherwise zero. |

## 3. Results

Table 2 provides the summary statistics of individual discount factors, group discount factors and other variables used in the analysis. The median individual (group) discount factors of fisheries, agrarian and urban societies are 0.100 (0.045), 0.500 (0.417) and 0.333 (0.278), respectively. These results reveal that both individual and group discount factors are the lowest (highest) in the fisheries (agrarian) society, and those in the urban society are in the middle. In other words, both individual and group discount factors non-monotonically change as societies transition from fisheries to agrarian and from agrarian to urban. Some researchers may claim a possibility of reverse causality in the sense that shortsighted (farsighted) people tend to be fishermen (farmers) reflecting the above results. However, this is not the case in our field experiments because migration among three societies is very low illustrated by the fact that fishermen, farmers and urban people have lived in 23, 21 and 20 years, in the same society, respectively and their fathers and grandfathers had been fishermen and farmers in fisheries and farming societies.

The results also show that the overall median (average) of group discount factors is 0.111 (0.353), while the overall median (average) of individual discount factors is 0.317 (0.414). This result indicates that group discount factors tend to be lower than individual discount factors. The median (average) group discount factors of group members with the lowest, middle and the highest discount factors are 0.040 (0.134), 0.100 (0.322) and 0.598 (0.556) in the fisheries society, 0.091 (0.184), 0.500 (0.505) and 0.909 (0.809) in the agrarian society, and 0.067 (0.154), 0.352 (0.397) and 0.727 (0.646) in the urban society, respectively. These results reflect the fact that individual discount factors in the fisheries society are consistently the lowest for every rank of individual discount factors in a group (the lowest, middle, and highest group members).

Regarding age, the overall average age of the subjects is 43 years. The average age of farmers is the highest because farmers tend to work longer than fishermen and urban people. This finding can be seen in the "max" row under age in Table 2, where the maximum age of farmers is 68 years. Moreover, the average ages of fishermen and urban people are not significantly different from each another since fishermen need to work in a labor-intensive manner and urban society attracts young people from rural areas to seek better jobs and opportunities. Table 2 also shows that the median household income is the highest (3.300) in the urban, the second-highest (3.100) in the agrarian and the lowest (2.500) in the fisheries societies. The income range is the widest in the urban society, which is consistent with the fact that the standard deviation (SD) of household income (2.773) in the urban society is the largest. This result reflects the fact that Jakarta is highly capitalistic and has a high income gap. The average number of household members is the largest (4.875) in the urban, the second-largest (4.485) in the fisheries and the lowest (4.222) in the agrarian societies, which reflects the fact that most farmers' children do not live with their parents since they usually move to urban areas for better jobs and opportunities. In summary, the individual and group discount factors in the fisheries society are consistently the lowest, and fishermen are relatively young and earn low incomes compared with those of farmers and urban people.

**Table 2.** Summary statistics of the field experiments and socioeconomic characteristics: 159 groups with 477 observations.

|  | Fisheries | Agrarian | Urban | Overall |
|---|---|---|---|---|
| **Group discount factor** | | | | |
| Median (Average) [1] | 0.045 (0.233) | 0.417 (0.452) | 0.278 (0.371) | 0.111 (0.353) |
| SD [2] | 0.335 | 0.383 | 0.347 | 0.366 |
| Min | 0.001 | 0.002 | 0.007 | 0.001 |
| Max | 0.952 | 0.952 | 0.952 | 0.952 |
| **Individual discount factor** | | | | |
| Median (Average) | 0.100 (0.337) | 0.500 (0.499) | 0.333 (0.399) | 0.317 (0.414) |
| SD | 0.360 | 0.373 | 0.331 | 0.362 |
| Min | 0.001 | 0.003 | 0.000 | 0.001 |
| Max | 0.870 | 0.952 | 0.833 | 0.952 |
| **Lowest individual discount factor** | | | | |
| Average (Median) [3] | 0.134 (0.040) | 0.184 (0.091) | 0.154 (0.067) | 0.158 (0.067) |
| SD | 0.233 | 0.237 | 0.188 | 0.222 |
| Min | 0.001 | 0.003 | 0.000 | 0.000 |
| Max | 0.952 | 0.952 | 0.952 | 0.952 |
| **Middle individual discount factor** | | | | |
| Average (Median) | 0.322 (0.100) | 0.505 (0.500) | 0.397 (0.352) | 0.410 (0.333) |
| SD | 0.339 | 0.344 | 0.289 | 0.333 |
| Min | 0.007 | 0.010 | 0.013 | 0.007 |
| Max | 0.952 | 0.952 | 0.952 | 0.952 |
| **Highest individual discount factor** | | | | |
| Average (Median) | 0.556 (0.598) | 0.809 (0.909) | 0.646 (0.727) | 0.674 (0.833) |
| SD | 0.365 | 0.221 | 0.301 | 0.317 |
| Min | 0.013 | 0.100 | 0.067 | 0.013 |
| Max | 0.952 | 0.952 | 0.952 | 0.952 |
| **Age** | | | | |
| Average (Median) | 40.839 (39.000) | 48.877 (48.333) | 40.250 (40.500) | 43.543 (43.667) |
| SD | 7.405 | 8.071 | 9.043 | 9.048 |
| Min | 30.000 | 29.667 | 23.000 | 23.000 |
| Max | 59.000 | 68.000 | 56.667 | 68.000 |
| **Household income** | | | | |
| Average (Median) | 2.777 (2.500) | 3.771 (3.100) | 4.253 (3.300) | 3.579 (3.000) |
| SD | 1.351 | 2.357 | 2.773 | 2.289 |
| Min | 1.167 | 1.733 | 1.100 | 1.100 |
| Max | 12.667 | 7.967 | 18.333 | 18.333 |
| **Number of household members** | | | | |
| Average (Median) | 4.485 (4.333) | 4.222 (4.000) | 4.875 (4.667) | 4.508 (4.333) |
| SD | 1.233 | 1.365 | 0.942 | 1.226 |
| Min | 2.000 | 2.333 | 3.333 | 2.000 |
| Max | 8.000 | 11.667 | 7.667 | 11.667 |
| **Number of proself members** | | | | |
| Average (Median) | 1.945 (2.000) | 1.719 (2.000) | 1.500 (1.000) | 1.730 (2.000) |
| SD | 0.897 | 0.940 | 0.923 | 0.932 |
| Min | 0.000 | 0.000 | 0.000 | 0.000 |
| Max | 3.000 | 3.000 | 3.000 | 3.000 |

[1] Average in parentheses for group and individual discount factors. [2] SD stands for standard deviation. [3] Median in parentheses for the variables other than the group and individual discount factors.

Figure 4a,b show the frequency distributions of the individual and group discount factors for fisheries, agrarian and urban societies. The vertical axis denotes the percentage of frequencies, and the horizontal axis denotes the discount factor. Regarding individual discount factors, Figure 4a demonstrates that the highest spike in the frequency distributions for the fisheries and urban societies occurs around 0, while the highest spike for the agrarian societies occurs around 1. On the other hand, Figure 4b shows that the highest spike in the frequency distribution of group discount factors occurs around 0 for every society, and the spikes in the fisheries society are higher than those in the agrarian and urban societies. These findings in the frequency distributions of individual and group discount factors across the three societies are in line with the summary statistics in Table 2. Based on the summary statistics, Figure 4a,b, we run a Mann-Whitney test to examine whether the distributions of the individual and group discount factors for any pair of fisheries, agrarian and urban societies are the same. The null hypothesis is that the distributions are independent of the three different societies. The results mostly reject the null hypothesis for individual (group) discount factors at the 1% (1%), 5% (1%) and 5% (18%) significance levels for fisheries vs. agrarian, fisheries vs. urban and agrarian vs. urban societies, respectively. Overall, the individual and group discount factors can be considered dependent on the three societies.

The summary statistics, frequency distributions and Mann-Whitney tests suggest that individual and group discount factors vary among the three societies. To further characterize the relationship between group and individual discount factors, we run betafit regression together with other independent variables. Table 3 presents the marginal effects of independent variables on the group discount factors with several model specifications. At first, we include only the agrarian and urban societies as dummy variables with fisheries as the reference in model 1 in Table 3 to account for possible concerns about effects of posttreatment variables. Previous literature explains that posttreatment effects occur as a bias for the estimates of treatment variables when any independent variable that get affected by the treatments is included in a regression together with treatment dummies [58]. Some researchers may claim a possibility of posttreatment effects in our research, we consider society dummy variables as treatments that might be considered to influence sociodemographic variables. In model 1, we find that society dummy variables are observed to be significant where a group discount factor in the fisheries society is 0.185 and 0.128 lower than the agrarian and urban societies, respectively.

We now exclude society dummy variables in model 2 in Table 3 to focus on examining how sociodemographic variables and the ranking of individual discount factors in a group affect group discount factors. The results show that group members with the lowest discount factors and middle discount factors and the number of proself members are statistically significant, playing important roles in determining the group discount factors. In addition, the number of household members influences group discount factors to a certain extent. (For the robustness check, we have tried to include household members as an independent variable in the regression, using the organization for economic cooperation and development (OECD) equivalence scale. While the OECD equivalence scale uses a criterion of children aged under 14, our data only contains the information about children aged under 12 within a household. This is because children in a fisheries society in Indonesia aged 12 graduate from elementary schools and they start to help their parents by working in fishery markets or other fisheries activities. Despite the difference in definition of children data between our research and OECD scaling, we have run a regression for robustness check using the OECD equivalence scale by considering the children aged under 12 as those aged under 14 in OECD scales, and we confirm that the results remain consistent and robust.) In particular, the results indicate that a group discount factor decreases by 0.0185 (0.0542) when the lowest (middle) individual discount factor in a group declines by 0.100. Likewise, a group discount factor decreases by 0.024 (0.036) with an increase in the number of household members (in the number of prosocial members in a group).

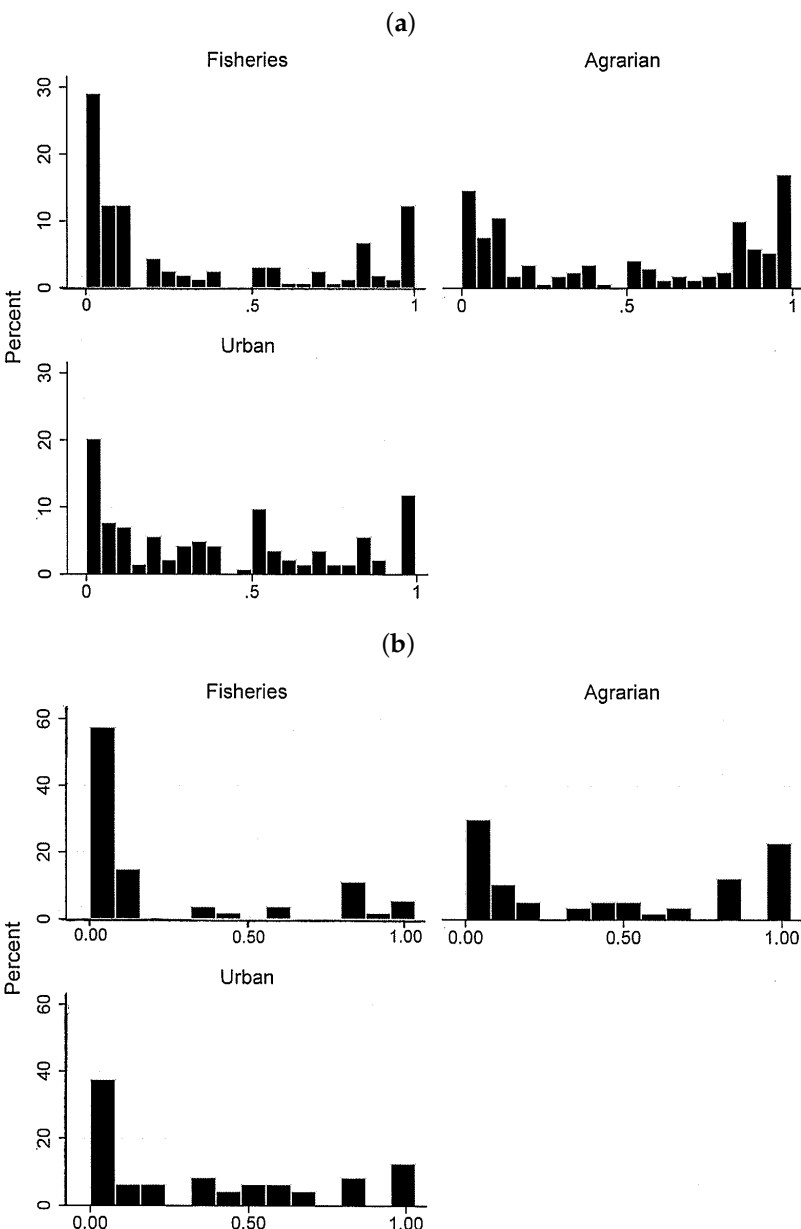

**Figure 4.** Frequency distributions of individual and group discount factors across the three societies. (**a**) Frequency distributions of individual discount factors across the three societies of fisheries, agrarian and urban; (**b**) Frequency distributions of group discount factors across the three societies of fisheries, agrarian and urban

**Table 3.** Marginal effects of individual discount factors on group discount factors in betafit regressions.

| Variables | Model 1 | Model 2 | Model 3 | Types of Societies | | |
| --- | --- | --- | --- | --- | --- | --- |
| | | | | Model 4 (Fisheries) | Model 5 (Agrarian) | Model 6 (Urban) |
| Individual discount factors | | | | | | |
| Lowest individual discount factor | | 0.185 ** | 0.208 *** | 0.277 *** | 0.193 | 0.208 |
| | | (0.074) | (0.075) | (0.095) | (0.150) | (0.183) |
| Middle individual discount factor | | 0.542 *** | 0.539 *** | 0.562 *** | 0.476 *** | 0.382 ** |
| | | (0.086) | (0.086) | (0.131) | (0.137) | (0.180) |
| Highest individual discount factor | | 0.075 | 0.052 | −0.098 | 0.249 | 0.253 |
| | | (0.068) | (0.068) | (0.073) | (0.153) | (0.174) |
| Age | | −0.001 | −0.002 | 0.000 | −0.006 | −0.001 |
| | | (0.002) | (0.002) | (0.003) | (0.005) | (0.004) |
| Household income | | 0.004 | 0.000 | 0.014 | 0.009 | 0.004 |
| | | (0.006) | (0.006) | (0.013) | (0.014) | (0.009) |
| Number of household members | | −0.024 * | −0.025 * | −0.010 | −0.055 ** | −0.011 |
| | | (0.014) | (0.014) | (0.014) | (0.023) | (0.025) |
| Number of proself members | | −0.036 * | −0.029 | −0.038 | −0.057 | 0.032 |
| | | (0.020) | (0.019) | (0.026) | (0.036) | (0.034) |
| Society dummy variables (The reference = the fisheries) | | | | | | |
| Agrarian | 0.185 *** | | 0.096 ** | | | |
| | (0.049) | | (0.049) | | | |
| Urban | 0.128 *** | | 0.086 * | | | |
| | (0.047) | | (0.046) | | | |
| Observations | 159 | 159 | 159 | 54 | 57 | 48 |

*** significant at 1 % level, ** significant at 5 % level and * significant at 10 % level.

To check the robustness of our results, we include all the independent variables as well as the agrarian and urban societies as dummy variables with fisheries as the reference in model 3, in addition to the baseline specification of model 2. Model 3 is estimated to examine how the transition of societies from fisheries to agrarian and from agrarian to urban may influence group time preferences. In model 3, the same qualitative results are observed as those in model 2, even with the agrarian and urban society dummy variables. The results in model 3 of Table 3 consistently show that a group discount factor decreases by 0.0208 (0.0539) when the lowest (middle) individual discount factor in the group declines by 0.100. An increase in several household members in a group leads to a 0.025 decrease in the group discount factor. Furthermore, the society dummy variables remain significant in the sense that a group discount factor in the agrarian (urban) society is likely to be 0.096 (0.086) higher than that in the fisheries society. This result shows that although we include society dummy variables in the regression, the lowest and middle discount factors of group members continue to affect the group discount factors, and the types of societies characterize group time preferences, which is consistent with Nguyen, and Johnson and Saunders [11,12].

We run separate regression models as models 4, 5 and 6 in Table 3 for fisheries, agrarian and urban societies, respectively, because we find that the farmer and urban dummy variables are significant in models 1 and 3, enabling us to examine whether the determinants of group discount factors differ across the three societies. Model 4 in Table 3 exhibits a qualitatively identical result with that of baseline specification in model 2, indicating that the group discount factor in the fisheries society decreases by 0.028 (0.056) when the lowest (middle) individual discount factor in a group declines by 0.100. On the other hand, the results in models 5 and 6 are similar in that the middle discount factor in a group is the only significant determinant. The result of model 5 in Table 3 shows that a group discount factor decreases by 0.048 when the middle discount factor in a group declines by 0.100 and that an increase in the number of household members decreases a group discount factor by 0.055. Finally, model 6 demonstrates that a group discount factor decreases by 0.038 when the middle individual discount factor in the group declines by 0.100.

Overall, our statistical analysis demonstrates that individual and group discount factors in the fisheries (agrarian) society are the lowest (highest), while those in the urban society are in the middle. Table 3 demonstrates that comparatively shortsighted people with the lowest and middle individual discount factors in a group remain consistently significant in models 2 and 3, while the society dummy variables are statistically and economically significant in models 1 and 3. The fisheries society (model 4) exhibits the same qualitative result as those of model 3 in that the lowest and middle individual discount factors play significant roles in determining group discount factors. The agrarian and urban societies (models 5 and 6) consistently show that the middle individual discount factor is the only significant variable characterizing the group discount factors. Although we have tried a variety of different regression specifications, our results in models 1–6 generally remain consistent and robust with respect to the roles of individual discount factors and society dummy variables in determining group discount factors. Some socioeconomic variables and other factors such as number of household members and number of proself members in a group are also identified to be statistically and economically significant depending on the specifications of the betafit regressions.

There are some possible explanations for our findings with respect to the roles of individual discount factors in determining group discount factors. First, fishermen in our study region (Karawang) are known to catch fish and earn income daily. They typically spend their entire daily income within that day and do not have motivation to save money for their future since they simply expect that they can continue fishing the next day to generate money for living. Additionally, most fishermen in the region believe that the fish stock is inexhaustible because God always provides fish in the sea (our questionnaire survey finds that 80.5 % of the fishermen hold this belief). Therefore, the daily life practices, the belief of an inexhaustible fish stock and their cultures make fishermen more shortsighted than farmers and urban people. This result is in line with the argument in Johnson and Saunders [11],

demonstrating that fishermen are more shortsighted than divers since divers are required to be patient to maintain healthy ocean and environmental conditions.

Fishermen in Karawang work in a fishing vessel as a group of 3 to 20 fishermen. In this environment, fishermen face two types of competition: intra-vessel and inter-vessel. In intra-vessel competitions, each fisherman in the same vessel has different types of tasks and job levels, competing to get promoted. On the one hand, inter-vessel competition occurs when a group of fishermen in a vessel compete with other groups in different vessels for better fishing spots and larger harvests. Carpenter and Seki, and Huang and Smith [59,60] illustrate that groups of fishermen compete to catch more fish and that the actions taken by groups of fishermen depend on the actions of other groups. Because the fishermen in our study region are under severe intra-vessel and inter-vessel competition, they become familiar with being or tend to be shortsighted at the individual and group levels in the way that comparatively shortsighted members in a group are more influential in determining the group discount factor.

Farmers in Karawang need to have patience and consideration for the future because farmers must wait six months for a series of cultivation and growth to harvest crops as one cycle. Moreover, they need to address substantial uncertainty. The major sources of uncertainty for farmers are natural disasters, which can destroy all agricultural production in a field. Although fishermen face the same type of risks and uncertainty from natural disasters, they can return to the sea and fish within a few days after a natural disaster. This is a fundamental difference between farmers and fishermen. In addition, farmers need to maintain their arable land for cultivation and harvesting since the land is their own property. Therefore, farmers in Karawang are motivated to save, invest, and accumulate capital and wealth by saving gold in preparation for an uncertain future. These daily practices and cultures appear to induce farmers to be patient or farsighted. Farmers typically work as a group to coordinate their efforts for irrigation, planting, growing, and harvesting to address uncertain climate conditions. For example, a group of farmers should cooperate, coordinate, and wait based on an irrigation schedule for fairness, avoiding the shortage of water among other groups of farmers. Overall, the aforementioned practices and cultures of the agrarian society in Karawang appear to induce farmers to be the most farsighted at the individual and group levels.

Finally, urban people in Jakarta usually live or work in an environment that is surrounded by technology and detached from nature. Urban people in Jakarta do not usually feel the limitations or constraints of basic needs, such as food, electricity, and water, whereas the fisheries and agrarian societies have some experience with tackling nature and feeling the limitations of various resources. In urban life in Jakarta, rice, meat, and fish are readily available in supermarkets and department stores, and such stores usually do not run short of any product because of national and international trade. In addition, by simply pressing a button, every energy source, such as electricity, becomes effective. This type of life implies that the basic needs of urban people tend to be readily available or become effective soon after their requests, which is not the case in fisheries and agrarian societies. On the other hand, urban people need to wait one month to receive salaries and need to study and improve themselves to become capable and competitive in the workplace of urban life. Therefore, urban life comes with a mixture of being shortsighted regarding basic needs and being farsighted regarding their career. Therefore, we conjecture that the individual and group discount factors in urban societies are in the middle between those of fisheries and agrarian societies.

Galor and Ozak [9] theorize that a production mode characterizes evolution of people's time preferences, deriving that an endowment (investment) mode of production induces people to evolve being shortsighted (farsighted). Following the definitions of production modes explained in Galor and Ozak [9], a fisheries (farming) society is considered to employ an endowment (investment) mode of production, while an urban society is considered to employ a mixture of both endowment and investment modes. With such interpretations of production modes in each society, our results follow their theory, because we find that individual and group discount factors both increase as societies transition from an endowment mode (fisheries) to an investment mode (agrarian) and then decrease as societies transition from an investment mode (agrarian) to a mixture of endowment and investment

modes (industrial one). Thus, the results in our experiments can be considered an important evidence to demonstrate and explain determinant factors and evolution of human time preferences by modes of production in societies at individual and group levels.

Another interesting finding is that comparatively shortsighted people (the lowest and middle) are more influential than farsighted people in determining group time preferences in models 2, 3 and 4. This result is in line, to a certain extent, with Ambrus et al., and He and Villeval [24,25], which elicit individual and group social preferences based on gift exchange, ultimatum and modified dictator games by asking subjects to allocate resources to themselves and others. To elicit individual social preferences, each subject plays a series of the games indicated above. To elicit group social preferences, a group of 5 members or 3 members is formed, and each group member is ranked with respect to social preferences based on his choices in the individual games [24,25]. Each group determines how to share resources between their groups and other groups. Ambrus et al., and He and Villeval [24,25] find that a member with the median social preference in a group has a significant effect on group social preferences because the highest and lowest subjects in a group tend to be attracted to the median member. In our case, however, the lowest individual discount factor is identified to be significant, which is different from Ambrus et al., and He and Villeval [24,25]. It is too early to conclude that the unique result in our analysis of group time preferences is generalizable; however, it may at least be the case that group time preferences are attracted to the relatively lower individual discount factors in a group.

In summary, our results reveal that individual and group discount factors non-monotonically change as societies transition, following the course of human history, through cultural and economic development. More specifically, individual and group discount factors both increase as societies transition from fisheries to agrarian and then decrease as societies transition from agrarian to industrial in that individual and group discount factors are the lowest (highest) in the fisheries (agrarian) society while those in the industrial society are in the middle. Our regression results also show that comparatively shortsighted people (the lowest and the middle) play important roles in characterizing group time preferences. These results can be considered to be important evidence of the factors influencing resource sustainability and economic development in each type of societies and of the further evolution of human time preferences in the future.

## 4. Conclusions

Previous research claims importance in considering the transition of societies from rural to urban to analyze social preferences and behaviors, demonstrating that people in urban societies are becoming more proself [2–5]. This paper considers three societies, namely, fisheries, farming and urban, as proxies of hunter-gatherer, agrarian and industrial societies, well representing the distinct cultures and daily practices that might shape human time preferences and behaviors. We have conducted a field experiment to elicit individual and group discount factors in the three societies of Indonesia. We find that individual and group discount factors are the lowest (highest) in the fisheries (agrarian) society, while those in the urban society are in the middle. We also find that the determinants of group discount factors differ across the three societies: members with the lowest and middle discount factors in a group play crucial roles in determining the group discount factor in the fisheries society while only the member with the middle discount factor is key in agrarian and urban societies. Overall, our results suggest that individual and group discount factors non-monotonically change as societies transition from fisheries to agrarian and from agrarian to urban and that comparatively shortsighted people (the lowest and middle) are more influential than farsighted people in determining group time preferences.

Finally, we note some limitations and possibilities for future studies. In this research, statistical analysis is the main tool used to characterize group time preferences through using the ranking of individual discount factors in a group. However, we have not examined the details of how group members determine or agree on group discount factors through their discussions in our field

experiments. If we use the qualitative-deliberative analysis from psychology on the transcribed group discussions, we should be able to identify how group members reach an agreement or compromise about group discount factors. If such an analysis is successfully conducted, we should be able to further clarify the detailed dynamic process of how people with the lowest or the middle discount factors in a group influence the group time preferences and to check the consistency with our statistical results. These caveats notwithstanding, it is our belief that this field experiment is an important first step to examining individual and group time preferences and their relation. Our results indicate that individual and group time preferences, as well as their determinants, evolve as societies change.

**Author Contributions:** Conceptualization, Y.H., K.K. and Y.K.; Data curation, Y.H. and K.K.; Formal analysis, Y.H., K.K. and Y.K.; Funding acquisition, K.K. and Y.K.; Investigation, Y.H., K.K. and Y.K.; Methodology, Y.H., K.K. and Y.K.; Project administration, Y.H. and K.K.; Resources, Y.H., K.K. and Y.K.; Software, Y.H. and K.K.; Supervision, K.K.; Validation, K.K. and Y.K.; Visualization, Y.H. and K.K.; Writing-original draft, Y.H. and K.K.; Writing-review and editing, Y.H., K.K. and Y.K.

**Funding:** The authors are grateful to the various supports from the Agency for Research and Human Resource of Marine Affairs and Fisheries, Ministry of Marine Affairs and Fisheries, Indonesia and the Grant-in-Aid for Challenging Exploratory Research (16K13354 and 16K13362) and Kochi University of Technology.

**Conflicts of Interest:** The authors declare no conflicts of interest.

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
