# Peer review of "Time Preferences between Individuals and Groups in the Transition from Hunter-Gatherer to Industrial Societies"

_sustainability, doi:10.3390/su11020395_

Round 1
Reviewer 1 Report
Summary: The authors conduct a field experiment in Indonesia to elicit individual and group discount factors across three societies (the hunter gatherers, the agrarian, and the industrial). They use fisheries, farming and urban societies, as proxies of the three societies and find that individual and group discount factors are the lowest in the fisheries, highest, in the agrarian, and medium in the urban. They also show that the determinants of group discount factors differ across societies: members of the lowest and middle discount factors in a group play an important role in determining the group discount factor in the fisheries society, while only the members with the middle discount factor are key in agrarian and urban societies. The results suggest that discount factors non-monotonically change as societies transition from fisheries to agrarian and from agrarian to urban and that relatively shortsighted people are more influential than farsighted people in determining group time preferences.
Comment: I enjoyed reading the paper. It addresses a very relevant topic. The experiment is properly designed, and the paper in general is clearly written.
Author Response
Please see the attachment for our replies to each comment made by referee 1.

Reviewer 2 Report
The paper analyses time preferences, both at individual and group basis, for three types of economies (fisheries, farming and urban). The analysis is based in field experiments (discounted elicitation experiment) applied in three Indonesia areas that represent the nature of the three societies. A social value orientation game is used to measure social preference and a betafit regression is employed for the empirical analysis of data
The article shows that the lowest discount factors appears in fisheries economics, the middle in urban economics and the highest in agrarian societies. Morover, group discount factors tend to be lower than individual discount discount factors. Furthermore, the determinants of group discount factors differ across the three societies; members with the lowest and middle discount factors in a group play an important role in making a group discount factor in fisheries societies, while only the member with the middle discount factor is key in agrarian and urban societies.
In general terms, the objective of the paper is well defined, the techniques used are appropriate and the results are unambiguous. It would be a valuable addition to the environmental economic literature. Therefore, I recommend publishing the article although I have some minor suggestions that may improve the paper.
1. The number of household members is one of the independent variable included in the beta regression analysis. Would not be better to include this variable using the OCDE equivalence scale? Do the results differ with this type of recalling?
2. I recommend including a reflexion about the role of non constant discount factor to represent time preferences in this type of societies. Some references for the use of non constant discount factors in fisheries economies are Ducan et al. (2011), Ekeland et al. (2015) and Da Rocha et al. (2018)
3. Typos: Axes labels in Figure 4 are skipped for the fisheries economies
References:
- Da Rocha, J.M., García-Cutrían, J., Gutiérrez, M.J., Touza, J., 2016. Reconciling yield stability with international fisheries agencies precautionary preferences: The role of non constant discount factorsin age structured models. Fish. Res. 173, 282–293
- Ducan, S., Hepburn, C., Papachristodoulou, A., 2011. Optimal harvesting of fish stocks under time-varying discount rate. J. Theor. Biol. 269, 166–173.
- Ekeland, I., Karp, L., Sumaila, R., 2015. Equilibrium resource management with altruistic overlapping generations. J. Environ. Econ. Manag. 70, 1–16.
Author Response
Please see the attachment for our replies to each comment made by referee 2.
